# Burden and Trends of Diet-Related Colorectal Cancer in OECD Countries: Systematic Analysis Based on Global Burden of Disease Study 1990–2021 with Projections to 2050

**DOI:** 10.3390/nu17081320

**Published:** 2025-04-10

**Authors:** Zegeye Abebe, Molla Mesele Wassie, Amy C. Reynolds, Yohannes Adama Melaku

**Affiliations:** 1Flinders Health and Medical Research Institute, College of Medicine and Public Health, Flinders University, Adelaide, SA 5042, Australia; molla.wassie@flinders.edu.au (M.M.W.); amy.reynolds@flinders.edu.au (A.C.R.); yohannes.melaku@flinders.edu.au (Y.A.M.); 2Department of Human Nutrition, Institute of Public Health, College of Medicine and Health Sciences, University of Gondar, Gondar P.O. Box 196, Ethiopia

**Keywords:** colorectal neoplasms, diet, disability-adjusted life years, death, global burden of disease

## Abstract

**Background**: An unhealthy diet is a major risk factor for colorectal cancer (CRC). This study assessed the diet-related CRC burden from 1990 to 2021 in Organisation for Economic Co-operation and Development (OECD) nations and estimated the burden until 2050. **Methods**: Data for OECD countries on diet-related CRC disability-adjusted life years (DALYs) and deaths were obtained from the Global Burden of Disease 2021 study. The estimated annual percent change (EAPC) was calculated to analyse the CRC burden attributable to dietary factors. A generalised additive model with a negative binomial distribution was used to predict the future burden of CRC attributable to dietary factors from 2021 to 2050. **Results**: In 2021, the age-standardised percentages of diet-related CRC DALYs and deaths were 39.1% (95% uncertainty interval (UI): 9.3, 61.3) and 39.0% (95% UI: 9.7, 60.9), respectively, in the OECD countries. Between 1990 and 2021, the age-standardised DALYs decreased from 185 to 129 per 100,000, and deaths decreased from 8 to 6 per 100,000 population for OECD countries. Similarly, the EAPC in the rates showed a downward trend (EAPC_deaths_ = −1.26 and EAPC_DALYs_ = −1.20). The estimated diet-related CRC DALYs and deaths are projected to increase to 4.1 million DALYs and 0.2 million deaths by 2050. There is a downward trend in CRC deaths (EAPC = 1.33 for both sexes) and in DALYs (−0.90 for males and −1.0 for females) from 1990 to 2050. **Conclusions**: The diet-related CRC burden remains significant. Implementing nutrition intervention programmes is necessary to promote access to affordable and nutritious foods and raise awareness about the importance of a healthy diet in reducing CRC risk.

## 1. Background

Colorectal cancer (CRC) remains a significant global burden, contributing to high levels of disability, death, and economic loss [1,2]. CRC is the fourth most common cancer and the second leading cause of cancer-related deaths worldwide, with nearly 1.9 million new cases and 935,000 deaths annually [3]. The incidence and mortality of CRC are rising each year and are expected to exceed 3.2 million new cases and 1.6 million deaths by 2040 [4]. In 2019, CRC contributed to 24.4 million disability-adjusted life years (DALYs) [5]. CRC is frequently diagnosed among individuals aged 50 years and older, with a higher prevalence among males [6].

While CRC affects individuals across Organisation for Economic Co-operation and Development (OECD) countries, there are notable variations in the incidence, mortality rates, and trends over time between nations [7]. Over the past decade, some OECD countries have experienced a gradual decrease in both the incidence and mortality of CRC [8,9]. This decline is attributed to advancements in healthcare, increased awareness about CRC risk factors including dietary factors, and the implementation of effective screening programmes [10]. However, in countries like Canada, while mortality rates have decreased, the CRC incidence has increased [11,12]. This discrepancy underscores the multifaceted nature of the CRC burden, which is influenced by population demographics, economic development, healthcare infrastructure, screening practices, and lifestyle factors such as diets [3,13,14].

Dietary behaviour is a significant risk factor contributing to the increasing prevalence of CRC worldwide [15]. Research indicates that the consumption of specific food items and nutrients such as a high intake of fruits, vegetables, fibre, and whole grains is associated with a reduced risk of CRC [16,17]. In contrast, diets high in red and processed meats, sweetened beverages, and alcohol have been shown to elevate the risk of developing CRC [18,19]. This evidence is supported by the latest dietary guidelines from the World Cancer Research Fund and the American Institute of Cancer Research (WCRF/AICR), which classify red and processed meats as probable and convincing carcinogenic risks for CRC, respectively [20]. Despite this evidence, recent studies on the diet-related CRC burdens in OECD countries are lacking.

A low intake of fruits, vegetables, whole grains, nuts and seeds, and foods high in sodium was identified as the leading dietary factor that contributed to the diet-related chronic disease burden among OECD countries, according to previous research from the Global Burden of Disease (GBD) 2015 study [21]. However, this existing work did not provide specific estimates of the diet-related CRC burdens. Also, the GBD 2021 study included 19,189 and 1176 new data sources for DALYs and risk factors, respectively, and new methods of dietary data analysis [22] were used. In addition, the number of OECD member countries increased from 35 in 2015 to 38 in 2021. Furthermore, no study has estimated the future diet-related CRC burdens in OECD countries. It is crucial to explore past trends and forecast the future trends of the diet-related CRC burden to design and implement effective prevention strategies. Therefore, this study aimed to determine the trends of the CRC burden attributed to dietary factors among OECD countries from 1990 to 2021 and projected these to 2050 using the GBD 2021 study.

## 2. Methods

### 2.1. Data Source

GBD data from 1990 to 2021 were analysed in this study, focusing on OECD member countries (*n* = 38): Australia, Austria, Belgium, Canada, Chile, Colombia, Costa Rica, the Czech Republic, Denmark, Estonia, Finland, France, Germany, Greece, Hungary, Iceland, Ireland, Israel, Italy, Japan, Latvia, Lithuania, Luxembourg, Mexico, the Netherlands, New Zealand, Norway, Poland, Portugal, Slovakia, Slovenia, South Korea, Spain, Sweden, Switzerland, Turkey, the United Kingdom, and the United States. Data on the diet-related CRC burden globally, at national and subnational levels, can be accessed through the GBD results tool portal: [https://collab2021.healthdata.org/gbd-results/ (accessed on 24 September 2024)].

### 2.2. GBD Overview

The GBD study periodically provides comprehensive estimates of global risk exposure and risk-related health burdens using all relevant available data. It estimates the incidence, prevalence, deaths, DALYs, years lived with disability, and years of life lost for 88 risk factors and 288 causes of death or injury. The estimates cover 204 countries and territories grouped into 21 regions and seven super-regions. It also includes subnational analyses for 21 countries and territories. Further details on the methods used have been published elsewhere [22,23].

### 2.3. Data Source and Processing

The GBD study collected dietary information from a variety of sources, including nationally and subnationally representative nutrition surveys that use the 24 h dietary recall methodology, food frequency questionnaires (FFQs), household budget surveys (HBSs), national sales data from Euromonitor (“sales”), and food availability data from the Food and Agriculture Organization of the United Nations (FAO) [24]. The input sources are accessible through an interactive citation tool available in the Global Health Data Exchange (GHDx): https://ghdx.healthdata.org/gbd-2021/sources (accessed on 24 September 2024).

The GBD study used the following steps to assess the exposure levels to all dietary risk factors [23]. They matched specific food items in the FAO data to dietary risk factors and estimated missing country–year data for each risk using spatiotemporal Gaussian process regression, with the log lag-distributed income per capita as the covariate. The GBD study used information about the consumption by age group and applied it to data sources that were not age-specific [23].

For each dietary risk factor, the GBD study estimated the global consumption pattern by age based on 24 h dietary recall surveys and applied consumption patterns to the all-ages data (FAO, sales, and HBSs). Then, the GBD study adjusted for bias using gold-standard data, which were 24 h dietary recall surveys. Other data sources, such as HBSs, FFQs, sales, and the FAO, were considered alternate data for the dietary intake and were aligned with the gold-standard data [23]. Out of the fifteen dietary factors reported in the GBD study, only six (diets low in whole grains, milk, fibre, and calcium, as well as diets high in red and processed meats) were determined to be CRC risk factors.

### 2.4. Estimating Risk Factors and Attributable Disease Burden

The GBD study followed the comparative risk assessment (CRA) framework to measure the risk–outcome association [24]. This first involved determining the effect size of the specific health outcomes occurring due to exposure to a particular risk factor. Following this, the levels and distribution of exposure to each risk factor were determined using two regression processes, spatiotemporal Gaussian process regression (ST-GPR) and disease model meta-regression (DisMod-MR 2.1) [23]. The ST-GPR employs regression methods that leverage spatial and temporal relationships to analyse a singular metric, such as the risk factor exposure or mortality rates, benefitting from collective strength across locations and time. Additionally, DisMod-MR serves as a Bayesian meta-regression tool, facilitating the comprehensive evaluation of all accessible data related to the incidence, prevalence, and mortality for a specific disease [25,26].

The GBD study generated relative risk (RR) estimates for the impact of the diet on CRC by analysing data from various studies, including randomised controlled trials, cohort studies, and case–control studies. These studies reported the RRs of mortality or morbidity from CRC based on dietary exposure. The data were obtained through systematic reviews [23].

The GBD 2021 study estimated the relationship between four food groups (whole grains, milk, red and processed meat) and two nutrients (fibre and calcium) and CRC burdens (DALYs and deaths). The GBD 2021 study used convincing or probable evidence on risk–outcome pairs and evaluated these using new burden-of-proof risk function (BPRF) analyses [23]. The BPRF accounted for unexplained between-study heterogeneity in the RR input data and provided conservative risk–outcome association.

The GBD study also established the levels of risk exposure that minimise the risk for the population, known as theoretical minimum risk exposure levels (TMRELs) [27]. The GBD study used various approaches to determine the dietary TMREL. For dietary risks that were strictly harmful, the TMREL was set at zero, while for protective dietary risks (except for vegetables), the GBD study utilised data from cohort studies used in the relative risk analysis to establish a TMREL. For each combination of a risk and outcome, the 85th percentile of the lower bound and the midpoints of all alternative (non-lowest) exposure ranges reported by the original study were calculated. Then, a weighted average across the outcomes for each lower bound and midpoint statistic was calculated, with the weights based on the total number of deaths associated with each outcome worldwide. Finally, a uniform distribution between these two values was created to serve as the TMREL [23]. The theoretical minimum risk exposure level for each dietary risk factor is provided in Appendix A. A similar but slightly different approach was taken for vegetables, as outlined in Stanaway et al., 2022 [23,28].

The proportion of attributable fractions quantifying the proportional change in health that would occur if risk exposure was reduced to the TMREL was independently computed for each risk–outcome pair with exposure estimates. Then, estimates of the burden attributed to the risk factor were computed. This was performed by quantifying the proportion of the disease burden, as measured by the product of the population attributable fraction (PAF) and the DALYs or deaths associated with the outcome, for each combination of age groups, sexes, locations, and years. The 95% uncertainty interval (UI) of the estimate was calculated as the 2.5th and 97.5th percentile values based on the mean values across 500 draws of the mean estimate [23].

### 2.5. Statistical Analysis

We collated the counts of deaths and DALYs, along with crude and age-standardised death rates (ASDRs), age-standardised rates (ASRs) of DALYs, and the annual rate of changes to assess the burden of diet-related CRC in OECD countries. The OECD countries were ranked from highest to lowest based on the age-standardised percentage of diet-related CRC DALYs and deaths.

The correlation between the estimated annual percentage of change (EAPC) and the socio-demographic index (SDI) was calculated to demonstrate the potential association between social and economic conditions influencing health and diet-related CRC in OECD countries. The SDI reflects the degree of socioeconomic development of countries and is a composite indicator of three factors: income per capita, educational attainment, and total fertility rate among people aged under 25 years [5]. The values of the SDI range from 0 to 1 and are further classified into five levels: high (≥0.80), high middle (≥0.69 and <0.80), middle (≥0.61 and <0.69), low middle (≥0.45 and <0.61), and low (<0.45) [5]. The EAPC was used to show changes in the trends of the CRC death and DALY rates from 1990 to 2021. The EAPC is equivalent to the annual change over a specific period and is calculated using the following formula [29]:
y=a+bx+ε

$$EAPC=100\%*(e^{\beta }-1)$$



y
 is the natural logarithm of the ASR, 
x
 is the calendar year, 
ε
 is the error term, and 
b
 signifies the positive or negative trend of the ASR within the linear formulation.

An ascending trend in the ASR is indicated when both the EAPC estimate and the 95% confidence intervals (CIs) are positive. In contrast, if both the EAPC estimates and 95% CIs are negative, the ASR has a downward trend. Otherwise, the ASR is stable [30].

The diet-related CRC burden was predicted up to 2050 using data from 1990 to 2019. The 2020 and 2021 data were excluded from the prediction due to the low reported number of cases, potentially due to COVID-19’s effect during that time. We assumed that the reported GBD estimates during this period might artificially affect the projections. Generalised additive models (GAMs) with a negative binomial family and log link with splines were used for the projection. We used the number of CRC deaths and DALYs attributed to dietary factors between 1990 and 2019 to project the future numbers of cases and rates. We also considered age, period, and cohort effects while building the GAMs [31]. The age and period were represented by the lower boundary of an age group and the calendar years of the GBD data released, respectively. Cohorts were calculated by subtracting the age from the period. Therefore, the GAMs took the following form:
$$y=\log\left(n\right)+s\left(a\right)+s\left(p\right)+s\left(c\right).$$


This model took the one-dimensional smooth functions for age, period, and cohort and used the penalised maximum likelihood method [31,32].

The diet-related CRC rates and number of deaths and DALYs were available by country from 1990 to 2021 and by 5-year age groups for both males and females aged 25–95+ years. In addition, the number exposed to risk for the ages of 25 to 95+ from 1990 to 2021 and the population projection from 2018 to 2050 were obtained from the GBD study. For age standardisation, the revised WHO world population weight was used as the standard population [33].

Let age be represented by 
a
, the period by 
p
, and the cohort by 
c
 (
c=p−a
). Let the person-years of exposure estimated according to the annual population be represented by 
n
. Moreover, let the predicted mean for the number of events be 
μ
. Let 
$wa_{a}$
 [34] be a set of positive-valued standard weights indexed over the age groups *a*, such that ∑*_a_w_a_* = 1. For a given period, *p*, the predicted age-standardised rate was calculated as follows [35]:
$${A{SR}_{P}=\Sigma }_{a}w_{a}\frac{\mu_{ap}}{n_{ap}}$$


To fit the rates of CRC deaths and DALYs attributed to dietary factors, the logs of the person-years of exposure were included as offsets in the regression model. The ASR denotes the event rate per 100,000 individuals, utilising a standardised global age structure, and is widely recognised as reflecting yearly rate variations over a specified timeframe. The projection was calculated for the OECD countries overall, and the sex-specific diet-related CRC incidence and mortality were projected to 2050 as the burden of CRC differs by sex. R version 4.3.0 with RStudio version 2023.6.0.421 was used to analyse the data, and we made projections using the R package ‘*mgcv*’ [36].

## 3. Results

### 3.1. Diet-Related CRC DALYs and Deaths in 2021

Detailed data on the diet-related CRC deaths and DALYs are summarised in Appendix A, respectively. In 2021, approximately 3.1 million or 39.1% (95% UI: 9.3, 61.3) of DALYs and 0.2 million or 39% (95% UI: 9.7, 60.9) of deaths were attributed to diet-related CRC. The proportion of diet-related CRC DALYs was slightly higher in those aged 50–75 years (39.3%; 95% UI: 8.7, 61.8) compared to those aged 25–49 years (39.0%; 95% UI: 8.9, 60.9) and 75 years and above (38.9%; 95% UI: 10.2, 60.4). However, the proportion of diet-related CRC deaths was higher among those aged 15–49 years (39.3%; 95% UI: 9.0, 61.4) compared to those aged 50–74 years (39.2%; 95% UI: 8.8, 61.7) and 75 and above (38.9; 95% UI: 10.4, 60.3) (Figure 1 and Figure 2).

For all age groups, the proportion of DALYs and deaths attributable to dietary risk factors was higher among females than males. However, the age-specific rate and number of CRC DALYs and deaths were significantly higher in males aged 50–79 years than in females (Appendix A).

In terms of the age-standardised percentages of deaths (ASPD) attributable to dietary risk among OECD countries, Chile (43.2%) ranked first, while Turkey (34.1%) ranked last (Appendix A). Similarly, the countries with the highest age-standardised percentages of diet-related CRC DALYs (ASP DALYs) showed similar patterns with CRC deaths. Chile had the highest percentage (42.9%), while Turkey had the lowest (33.8%) (Appendix A).

### 3.2. Trends of Diet-Related CRC DALYs and Deaths Between 1990 and 2021

The total cases of diet-related CRC deaths increased steadily from 111,928 (95% UI: 27,640, 173,390) in 1990 to 154,543 (95% UI: 37,764, 243,081) in 2021, with an increase of 38.1%. However, the ASDR decreased from 10 per 100,000 population (95% UI: 2, 15) to 7 per 100,000 population (95% UI: 1, 11), with an EAPC = −1.26 (95% CI: −1.29, −1.23) (Table 1).

The proportion of diet-related CRC DALYs and deaths among young adults showed an increasing trend from 1990. In 2021, a poor diet contributed to 39.3% (95% UI: 8.9, 61.4) of CRC deaths compared to 38.9% (95% UI: 9.2, 60.6) in 1990 in young adults (aged less than 50 years). A poor diet contributed to 39.2% (95% UI: 8.8, 61.7) deaths in middle age (50–74 years) compared to 39.3% (95% UI: 9.0, 61.8) in 1990, indicating a decrease in the diet-related CRC burden among this age group (Figure 3).

However, the rate of diet-related CRC DALYs and deaths was higher among males. It was found that the diet-related rate of CRC deaths was higher among those aged 75 years and older (88 and 74 deaths per 100,000 among males and females, respectively) (Figure 4).

Since 1990, three OECD countries have experienced an annual increase in the ASDR, with the fastest increase being in Costa Rica (EAPC = 1.67; 95% CI: 1.48, 1.86), followed by Mexico (EAPC = 1.03; 95% CI: 0.87, 1.19), Chile (EAPC = 0.48; 95% CI: 0.35, 0.61), and Poland (EAPC = 0.21; 95% CI: 0.04, 0.38). In addition, Colombia (EAPC = 0.09; 95% CI: −0.08, 0.27) and Lithuania (EAPC = −0.05, 95% CI: −0.21, 0.10) demonstrated a consistent trend over the last three decades, showing that there has been no change in the diet-related ASDR. All other countries had negative trends over time, indicating a decreasing trend in diet-related CRC deaths (Table 1 and Appendix A).

Similarly, the number of DALYs attributed to all dietary risks for CRC was increased from 2,447,568 (95% UI: 587,458, 3,793,421) in 1990 to 3,088,502 (95% UI: 722,609, 4,844,604) in 2021, with an increase of 26.2%. The ASP of DALYs attributed to dietary risks remained the same between 1990 and 2021. However, the ASR of DALYs decreased from 185 (95% UI: 44, 286) in 1990 to 129 (95% UI: 30, 202) in 2021. The EAPC in the ASR also showed a downward trend (EAPC = −1.2; 95% CI: −1.24, −1.18). The EAPC in the ASR of DALYs attributed to all dietary factors in CRC had a similar trend to that of deaths attributed to dietary factors among OECD countries. Over the past three decades, four OECD countries showed an increase in DALY rates, with Costa Rica (EAPC = 1.85; 95% CI: 1.63, 2.06) having the highest increase, followed by Mexico (EAPC = 1.47; 95% CI: 1.33, 1.61), Chile (EAPC = 0.57; 95% CI: 0.43, 0.71), and Colombia (EAPC = 0.31; 95% CI: 0.13, 0.49). Meanwhile, Poland (EAPC = 0.07; 95% CI: −0.1, 0.24) showed a consistent trend over the last 31 years. All other OECD countries experienced a downward trend over the same period, with the most significant decrease observed in Austria (EAPC = −2.44, 95% CI: −2.49, −2.39) (Table 2 and Appendix A). From 1990 to 2021, the DALYs and ASDR remained higher in males than in females.

### 3.3. CRC Death and DALY Burden Attributed to Each Dietary Risk Factor

Across OECD countries, diets low in whole grains and milk and a diet high in red meat were the top three contributing factors to diet-related CRC deaths and DALYs (Figure 5 and Figure 6). In 2021, approximately 18.0% (95% UI: 7.53, 26.82) and 18.02% (95% UI: 7.55, 26.84) of DALYs and deaths were linked to low whole grain intakes, respectively (Appendix A). The trends of each diet-related factor of CRC DALYs and deaths were consistently the same over the last decades (Figure 7). The trends of each dietary risk factor in every country can be found in Appendix A.

### 3.4. Relationship Between Socio–Demographic Index and Estimated Annual Percentage of Change in Diet–Related CRC Burdens

Figure 8 illustrates the relationship between the SDI and EAPC. There was a significant inverse relationship between the SDI and the EAPC in the ASDR, evident in both 1990 and 2021, with correlation coefficients of −0.46 (*p* = 0.004) and −0.59 (*p* < 0.0001), respectively. Likewise, the correlation between the SDI and EAPC and the ASR of DALYs, also in 1990 and 2021, was negative, exhibiting coefficients of −0.41 (*p* = 0.01) and −0.64 (*p* < 0.001), respectively, indicating a decrease in CRC burdens with an increase in the SDI value.

### 3.5. Diet-Related CRC DALY and Death Burden Projection

Figure 9 shows both the change in the trends of the CRC DALY and death burdens attributed to dietary factors in OECD countries and the projections through to 2050. The number of CRC deaths attributed to dietary factors is projected to be 214,049 in 2050, a 39.9% increase from 2019 to 2050. The number of CRC DALYs is also projected to be 4,098,672 in 2050, a similar increase of 32.9% from 2019 to 2050. The number of DALYs and deaths is predicted to be higher in males than females. For males, the predicted ASDR is 5.69 per 100,000 in 2050, which is a decrease of 8.4% compared to 2019. The ASDR among females is predicted to be 4.0 per 100,000, a decrease of 18.4% relative to 2019.

For the predicted DALYs in 2050, the ASR of DALYs among males is predicted to be 151 per 100,000 with a decrease of 5.6%, and among females, it is predicted to be 90 per 100,000, with a similar 17.8% decrease. The EAPC also indicates a downward trend in CRC deaths (EAPC = −1.33 and 95% CI of −1.36 to −1.30 for males; EAPC = −1.33 and 95% CI of −1.37 to −1.29 for females) and DALYs (APC = −0.90 and 95% CI of −0.98 to −0.83 for males; EAPC = −0.95 and 95% CI of −1.00 to −0.89 for females) attributed to dietary factors from 1990 to 2050.

## 4. Discussion

### 4.1. Summary of Findings

This study aimed to explore the trends and burden of diet-related CRC in OECD countries using GBD data between 1990 and 2021. The number of diet-related CRC deaths and DALYs has increased over the last three decades, with men showing higher death and DALY burdens than women. However, the overall trends of the ASR of DALYs and ASDR decreased from 1990 to 2021. A diet low in whole grains, high in red meat, low in milk, high in processed meat, low in calcium, and low in fibre contributes to the diet-related burdens of CRC. Projections suggest that by 2050, CRC deaths and DALYs attributed to dietary factors will increase by 39.9% and 32.9%, respectively. Nevertheless, since 1990, the ASR of DALYs and ASDR have been declining and are expected to decrease in OECD countries. This study also indicated a clear pattern of decreasing diet-related CRC burdens with rising SDI levels over time across OECD countries.

These findings suggest that despite a decline in the ASDR and DALYs, the burden of diet-related CRC remains high in OECD countries. In 2021, the burden of diet-related CRC was significant in OECD countries, with 154,543 deaths (39%; 95% UI: 9.7, 60.9) and 3,088,502 DALYs (39.1%; 95% UI: 9.3, 61.3). These figures represented a substantial portion of the total CRC burden, accounting for more than 52.5% of all CRC deaths and 52.8% of all CRC DALYs in OECD countries. This underscores the critical role of dietary intake as a modifiable behavioural factor, closely linked to the burden of CRC.

The study findings also reveal that the burden of diet-related CRC deaths and DALYs is higher in males compared to females. This observation aligns with previous studies indicating that males are more inclined to consume diets characterised by a lower intake of whole grains and fibre [37,38] and a higher consumption of red and processed meats [39,40,41]. These dietary patterns among males likely contribute to the increased CRC risk and associated morbidity. The consumption of diets low in fibre and whole grains, which are essential components of a healthy diet, may result in decreased bowel motility and increased exposure to potentially carcinogenic substances in the colon [42,43]. Additionally, a low intake of dietary fibre, combined with a high intake of processed foods, can lead to the dysbiosis of the gut microbiome, which may increase the risk of CRC [44]. Conversely, a higher intake of red and processed meats is associated with elevated levels of saturated fats, heme iron, and other compounds that have been linked to an increased risk of CRC [45].

CRC, traditionally associated with individuals aged 50 years and older, is witnessing an emerging trend for an increased prevalence among younger adults (those aged below 50 years) [46]. The reason for the shift in demographics has been linked to changes in dietary habits and lifestyle factors [47,48]. The period from 1990 to 2021 saw a significant increase in the burden of diet-related CRC among younger adults, which is thought to have been driven by various factors. One possible factor behind this trend could be the shift from fresh and nutritious diets to processed and unhealthy ones among young adults [49]. Unhealthy eating habits, including a higher consumption of alcohol and processed meat, have become increasingly prevalent [50]. These dietary changes, coupled with sedentary lifestyles, may be playing a role in the observed rise in the CRC incidence [48].

Moreover, the escalating rates of overweight and obesity among young adults are believed to be influencing the upward trend of CRC [51]. The correlation between excess body weight and an increased risk of developing CRC underscores the significance of addressing this aspect in the context of public health interventions [52]. Another noteworthy aspect is the current focus of CRC screening programmes being exclusively on older adults, neglecting the younger population [53,54]. This oversight may contribute to the rise in the diet-related burdens of CRC among young adults. Expanding the scope of CRC screening to include younger age groups could potentially lead to early detection and intervention, thereby mitigating the increasing CRC burden in this demographic [55].

Over the past 31 years, the percentage of change in the ASDR and DALY rate of CRC has exhibited a downward trend in most OECD countries, suggestive of progress in addressing diet-related CRC burdens. However, exceptions to this trend have been observed in Chile, Costa Rica, Mexico, Poland, and Colombia, where diet-related CRC deaths and DALYs have not shown the same decline, and require further attention. These findings align with the overall CRC burdens in the above countries. In Chile, Costa Rica, and Colombia, the overall incidence rate of CRC has increased by an average of 1.3% to 4.1% per year [56]. In addition, the observed increase in the trends of CRC burdens may reflect unhealthy dietary patterns [57]. Among OECD countries, Chile, Mexico, and Colombia have the highest red and processed meat consumption [57], which is associated with the increased risk and persistence of CRC burdens [45,58].

In this study, it was found that the EAPC decreased as the SDI increased. This means that countries with higher SDI values experienced a rapid decrease in diet-related CRC burdens from 1990 to 2021. This observed declining trend may be attributed to the fact that countries with higher SDI scores are more likely to have comprehensive CRC screening programmes in place, as well as well-established interventions aimed at lowering CRC mortality rates [53,54]. Conversely, countries with SDI scores ranging from low to high–middle have undergone a dietary shift from traditional, minimally processed meals to a more Westernised diet, contributing to an increase in the burden of diet-related CRC [59]. The other potential reason is that countries with a higher SDI usually have more effective public health initiatives and educational programmes that promote healthy eating habits in comparison to lower-SDI regions [60]. It is important to address these disparities through targeted interventions and education among low-SDI countries.

### 4.2. Implications of Findings

This study highlights the importance of addressing dietary intake and nutritional choices in efforts to reduce the global burden of CRC [61]. Designing and tailoring healthy policies, dietary guidelines, public awareness campaigns, and healthcare services to effectively improve the burden attributable to dietary intake is of importance [1,62].

Our study supports the recommendations outlined in the joint guidelines from the WCRF and AICR [63], which advocate for augmenting the whole grain intake and reducing the intake of red and processed meat as a means to mitigate the CRC risk [63]. Moreover, these findings are in line with a dose–response meta-analysis of six observational studies on whole grain consumption and the CRC risk, which reported a 17% reduction in the CRC risk associated with each incremental increase of 90× *g* in the daily whole grain consumption [64]. These results further underscore the potential preventive benefits of whole grain-rich diets against CRC incidence. Consistent with our observations, previous GBD studies [65] and umbrella reviews of meta-analyses [66] have highlighted the adverse impact of diets low in fibre and milk while being high in red meat on CRC burdens. This convergence of evidence underscores the significance of dietary choices in influencing CRC outcomes and reinforces the importance of adopting dietary patterns aligned with recommendations aimed at reducing the CRC risk [63].

The observed gender disparity in dietary habits and the CRC burden mirrors similar trends observed in the association between dietary factors and other health outcomes. For instance, studies have shown that males tend to have higher rates of cardiovascular disease [67] and certain types of cancer [68], which can also be influenced by dietary choices. Therefore, addressing gender-specific dietary patterns and promoting healthier eating habits is crucial not only for reducing the burden of CRC but also for improving overall health outcomes [69].

The observed increase in the overall trends in some of the OCED countries may be attributed to the rise in obesity [70], physical inactivity [71,72], alcohol consumption [73], and low consumption of vegetables and fruits in diets [74,75]. Addressing the persistent burden of CRC in these countries requires targeted interventions aimed at promoting healthier dietary habits [76]. Public health initiatives focusing on reducing red and processed meat consumption, promoting balanced diets rich in fruits, vegetables, and whole grains, and raising awareness about the link between the diet and CRC risk are essential [77]. Interventions targeting other behavioural risk factors at the population level such as smoking cessation, obesity prevention strategies, and the limitation of alcoholic drinks may help in reducing the CRC burdens [63], particularly in countries where risks do not appear to have improved. In addition, dietary intake screening along with CRC screening for those at risk of CRC may provide additional advantages [78].

Given the current burden of and projected increase in CRC, a comprehensive approach is essential. First, ensuring access to healthy food options is crucial for promoting better nutrition across diverse populations. This approach to broad population-level interventions may improve overall health as well as reduce the burden of CRC. Second, OECD countries can learn from successful interventions in specific nations, such as implementing taxes on sugar-sweetened beverages or regulations regarding processed red and meat consumption, to mitigate CRC burdens [79]. Third, price reduction strategies should be adopted for healthy whole grains, fruits, vegetables, and less processed food items to promote healthy eating habits [80]. Furthermore, due to the high burdens of diet-related CRC trends, particularly noting the elevated CRC incidence in countries with a high consumption of processed foods and red meat [57], OECD countries should emphasise specific recommendations within the WCRF dietary guidelines [63]. These recommendations include reducing processed meat consumption, limiting red meat intake, and promoting diets rich in fruits, vegetables, and whole grains to alleviate CRC burdens [63]. Finally, increasing the awareness of the diet–CRC connection and offering practical guidance for dietary changes can aid in reducing CRC burdens [78].

### 4.3. The Limitations of the Study

While our analysis provides valuable insights into the burdens of CRC attributed to dietary factors in OECD countries, it is essential to acknowledge certain limitations inherent in our approach. Firstly, our report primarily focused on four food items (whole grains, red and processed meat, and milk) and two nutrients (fibre and calcium). However, in reality, the complexity of human dietary patterns necessitates the consideration of a broader array of food items and nutrients to comprehensively understand their collective influence on CRC burdens, as the interactions among dietary components can be intricate and potentially synergistic [81]. The singular focus on individual food items and nutrients in the GBD study may not fully capture the intricate interplay among dietary components, which could be vital in elucidating their combined effects on CRC burdens. Secondly, while whole grains are indeed rich in fibre, the inclusion of whole grains as a separate dietary factor alongside fibre may lead to them overlapping when estimating the risk. This overlap underscores the challenge of separating the individual contributions of specific dietary components when they are inherently linked, as is the case with whole grains and fibre. Third, the results for the GBD 2020 and 2021 studies were collected during the COVID-19 pandemic. The impact of COVID-19 on the healthcare system may have caused delays in cancer screenings, diagnoses, and treatments. This could have led to underreporting or missed cases, potentially skewing the results related to diet-related CRC burdens and historical trends. Finally, our projections for the diet-related burden of CRC in 2050 were based on four food groups and two nutrients. Additionally, dietary intakes are dynamic and influenced by various factors, including socioeconomic changes, food policies, technological advancements in food production, and public health initiatives. As a result, our estimates may either underestimate or overestimate future CRC burdens due to changing dietary trends.

## 5. Conclusions

Diet-related CRC is a significant issue among OECD countries. Although the EAPC for deaths and DALYs has generally decreased across these nations, some countries, such as Chile, Costa Rica, Mexico, Poland, and Colombia, are experiencing rising trends that pose substantial public health challenges. The burden of CRC attributed to dietary risks is typically higher in males, and this gender difference has become more pronounced over time. Additionally, the proportion of diet-related CRC cases among younger adults is on the rise. Based on our projections, despite a decline in the ASDR and ASR of DALYs, the total number of diet-related CRC deaths and DALYs may continue to increase until 2050. Public health initiatives focusing on increasing awareness about the importance of a diet rich in fibre, whole grains, fruits, and vegetables, while reducing the consumption of red and processed meats, can play a significant role in mitigating the burden of CRC. It is important to consider how these interventions need to be tailored to different countries and facilitated or supported in order to be effective.

## Figures and Tables

**Figure 1 nutrients-17-01320-f001:**
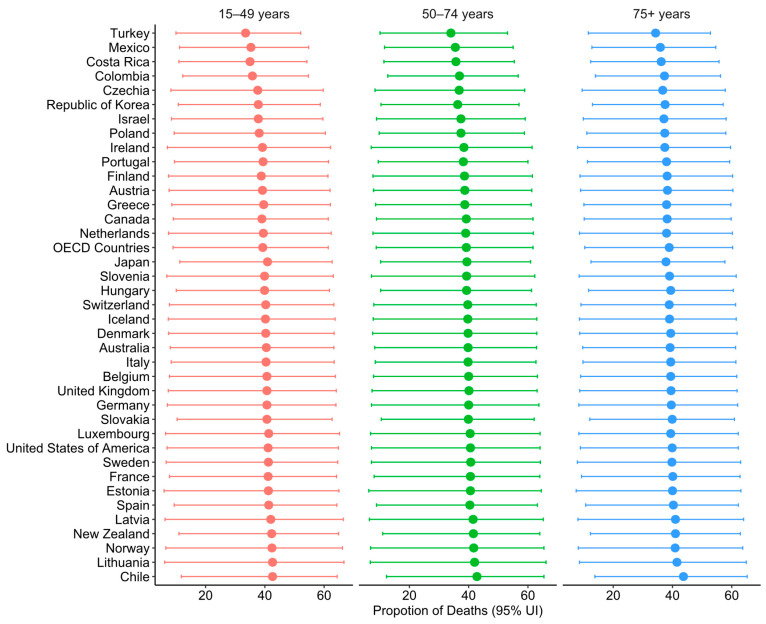
Proportion of diet-related CRC deaths by age group in OECD countries in 2021.

**Figure 2 nutrients-17-01320-f002:**
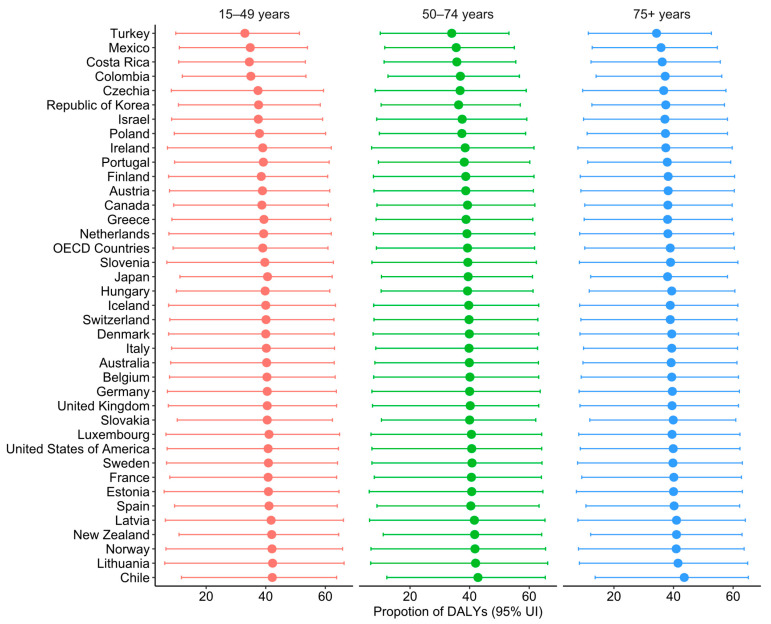
Proportion of diet-related CRC DALYs by age group in OECD countries in 2021.

**Figure 3 nutrients-17-01320-f003:**
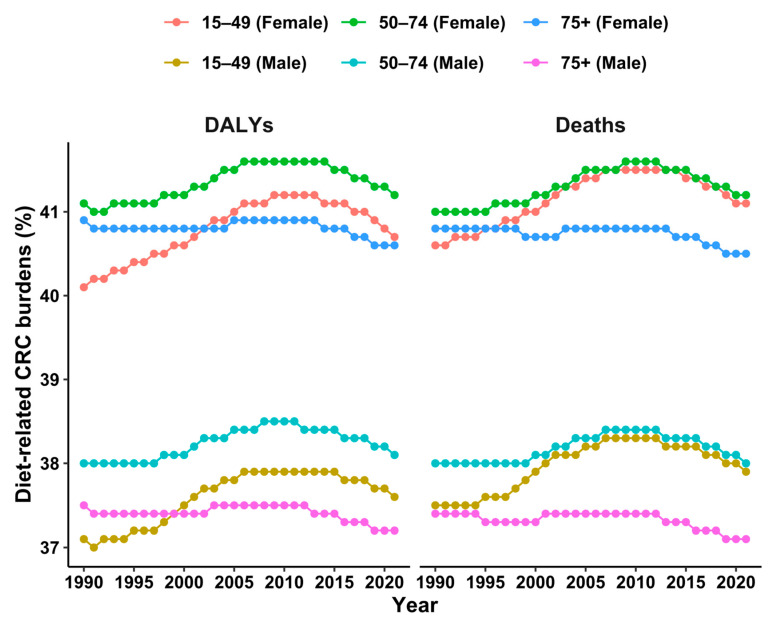
Diet-related CRC burden by sex and age group from 1990 to 2021.

**Figure 4 nutrients-17-01320-f004:**
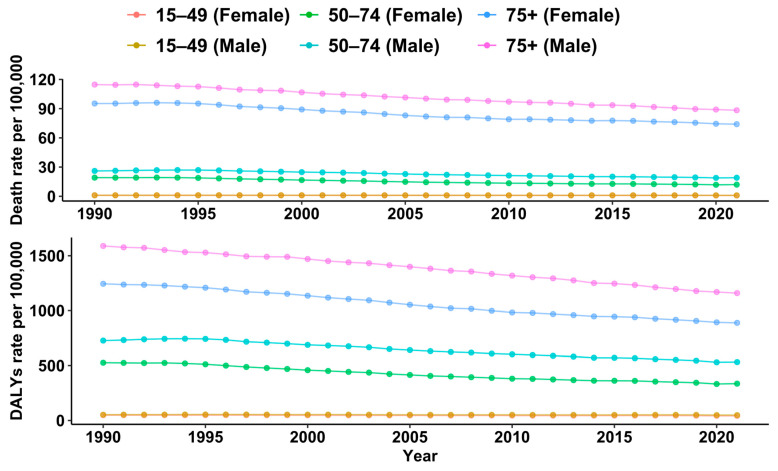
Diet-related rate of CRC burden by sex and age group from 1990 to 2021. Note: Estimate for 15–49 age group overlaps and line overlaps in Figure 4.

**Figure 5 nutrients-17-01320-f005:**
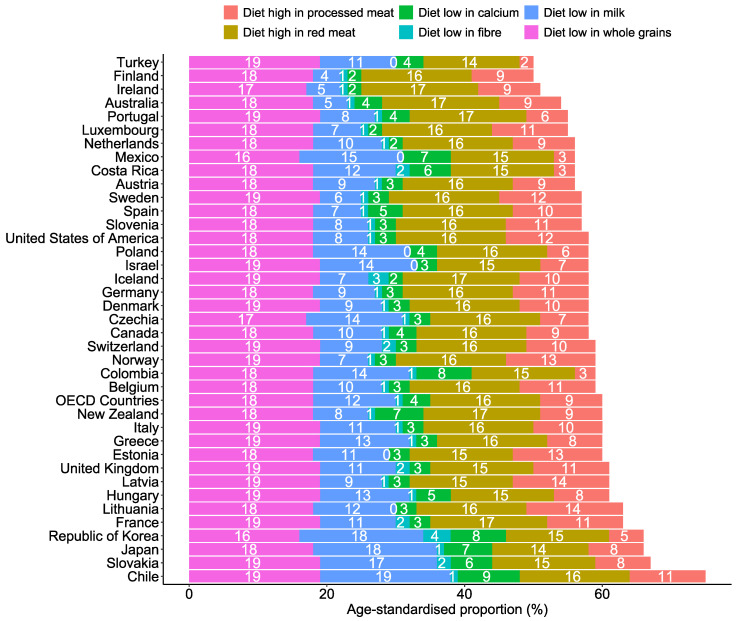
Specific diet–related age-standardised proportion of CRC deaths among OECD countries in 2021.

**Figure 6 nutrients-17-01320-f006:**
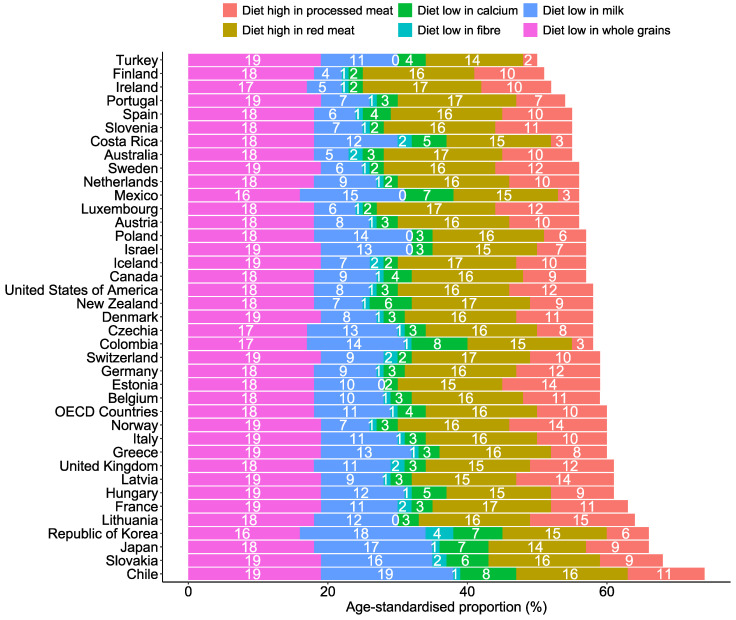
Specific diet–related age-standardised proportion of CRC DALYs among OECD countries in 2021.

**Figure 7 nutrients-17-01320-f007:**
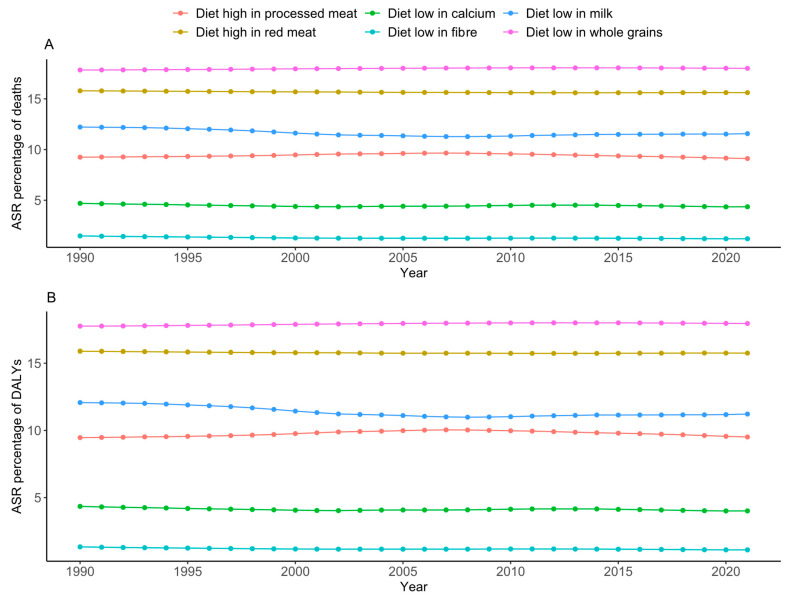
Trends of specific diet–related CRC DALYs and deaths among OECD countries from 1990 to 2021. (**A**) Age-standardised percentage of CRC deaths, (**B**) age-standardised percentage of CRC DALYs.

**Figure 8 nutrients-17-01320-f008:**
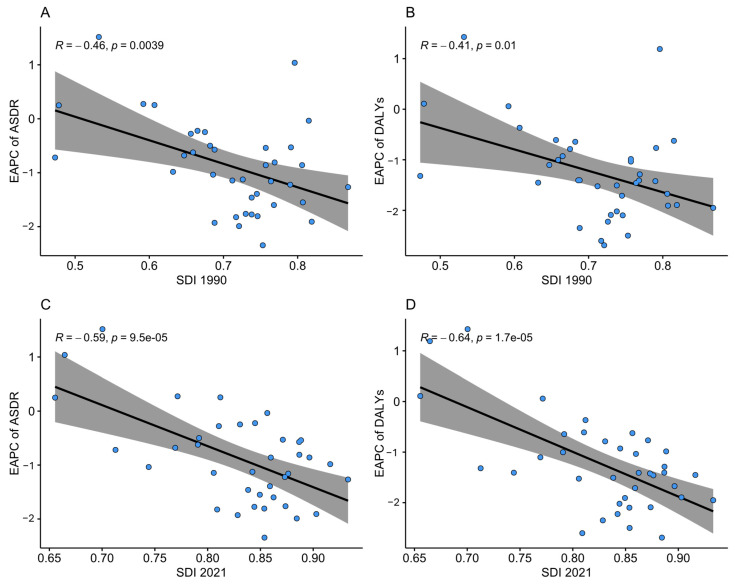
Correlation between SDI and EAPC from 1990 to 2019. (**A**) Correlation between EAPC in ASDR and SDI in 1990, (**B**) correlation between EAPC in ASR of DALYs and SDI in 1990, (**C**) correlation between EAPC in ASDR and SDI in 2021, and (**D**) correlation between EAPC in ASR of DALYs and SDI in 2021. Note: Solid lines represent direction of association, and shaded areas represent 95% confidence interval. Dots indicate SDI values of each country. R represents Spearman correlation coefficient and *p* values indicate level of statistical significance. Abbreviations: ASDR = age-standardised death rate; DALYs = disability-adjusted life years; EAPC = estimated annual percentage of change, SDI = socio-demographic index.

**Figure 9 nutrients-17-01320-f009:**
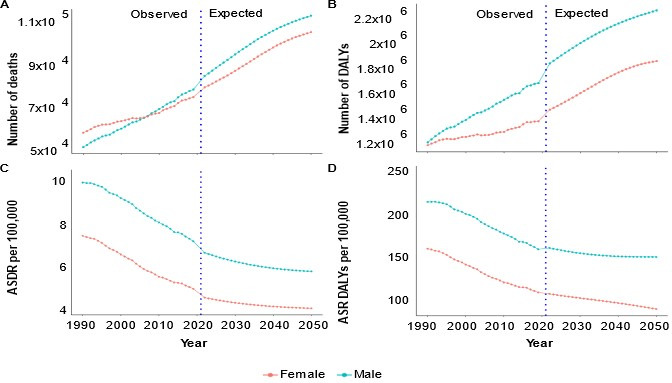
Prediction of age-standardised rate and number of CRC DALYs and deaths attributed to dietary factors in OECD countries. (**A**) Number of CRC deaths, (**B**) number of CRC DALYs, (**C**) ASDR per 100,000 from 1990 to 2050, and (**D**) ASR of DALYs per 100,000 from 1990 to 2050. Abbreviations: ASDR = age-standardised death rate; ASR = age-standardised rate; DALYs = disability-adjusted life years.

**Table 1 nutrients-17-01320-t001:** Death, ASDR, **ASPD,** and change trends of CRC attributed to dietary risk between 1990 and 2021 among OECD countries.

**Country**	**Number (95% UI)**		**ASPD (95% UI)**		**EAPC in ASPD (95% CI)**	**ASDR (95% UI)**		**EAPC in ASDR (95% CI)**
	**1990**	**2021**	**1990**	**2021**	**1990** **–** **2021**	**1990**	**2021**	**1990** **–** **2021**
OECD	111,929 (27,640, 173,390)	154,543 (37,765, 243,081)	39.4 (9.7, 61.4)	39.1 (9.5, 61.0)	−0.01 (−0.02, 0)	8 (2, 13)	6 (1, 9)	−1.26 (−1.29, −1.23)
Australia	1806 (437, 2869)	2638 (606, 4237)	39.2 (9.4, 61.7)	39.5 (9.1, 62.1)	0.03 (0.02, 0.04)	9 (2, 15)	6 (1, 9)	−1.9 (−2.0, −1.8)
Austria	1167 (283, 1855)	903 (192, 1451)	38.6 (9.3, 60.3)	38.4 (8.6, 60.8)	0.01 (−0.001, 0.01)	10 (2, 15)	5 (1, 7)	−2.42 (−2,47, −2.36)
Belgium	1573 (376, 2454)	1459 (301, 2332)	39.3 (9.2, 61.7)	39.7 (8.6, 62.3)	0.07 (0.06, 0.08)	10 (2, 16)	6 (1, 9)	−1.73 (−1.83, −1.64)
Canada	2637 (702, 4122)	4159 (989, 6645)	39.2 (10.7, 61.0)	38.6 (9.7, 60.7)	−0.04 (−0.05, −0.37)	8 (2, 13)	5 (1, 9)	−1.08 (−1.19, −0.98)
Chile	515 (183, 769)	1476 (428, 2274)	43.6 (15.3, 64.5)	43.2 (13, 65.4)	−0.01 (−0.01, 0.03)	5 (2, 8)	6 (2, 9)	0.48 (0.35, 0.61)
Colombia	602 (242, 880)	2142 (720, 3374)	38.6 (15.7, 57.2)	37.0 (13.2, 56.3)	−0.10 (−0.14, −0.06)	4 (1, 5)	4 (1, 6)	0.09 −0.08, 0.27)
Costa Rica	60 (21, 90)	299 (95, 473)	37.1 (13.2, 56.4)	35.9 (11.8, 55.4)	−0.13 (−0.14, −0.12)	4 (1, 5)	5 (2, 9)	1.67 (1.48, 1.86)
Czechia	1784 (459, 2891)	1663 (417, 2699)	37.0 (9.7, 58.5)	36.8 (8.9, 58.4)	−0.02 (−0.03, −0.01)	13 (3, 21)	7 (2, 12)	−2.07 (−2.26, −1.89)
Denmark	773 (181, 1233)	969 (194, 1558)	39.2 (8.9, 61.5)	39.5 (8.3, 62.3)	0.04 (0.03, 0.05)	9 (2, 15)	8 (1, 12)	−1.02 (−1.31, −0.74)
Estonia	156 (27, 247)	213 (34, 348)	41.3 (7.6, 64.7)	40.3 (7, 63.7)	−0.08 (−0.1, −0.07)	8 (1, 12)	7 (1, 12)	−0.49 (−0.65, −0.33)
Finland	421 (98, 688)	608 (126, 991)	38.8 (8.8, 60.9)	38.4 (8.4, 60.9)	−0.02 (−0.03, −0.01)	6 (1, 9)	4 (1, 7)	−0.92 (−1.03, −0.82)
France	8375 (1917, 13,152)	9905 (2054, 16,027)	40.1 (8.9, 63.0)	40.3 (8.8, 63.2)	0.025 (0.02, 0.03)	10 (2, 15)	6 (1, 10)	−1.44 (−1.49, −1.39)
Germany	13,923 (3089, 21,858)	12,305 (2324, 19,763)	40.0 (8.8, 62.5)	39.7 (7.9, 62.7)	−0.015 (−0.02, −01)	11 (2, 17)	6 (1, 9)	−2.19 (−2.3, −2.08)
Greece	909 (231, 1429)	1485 (348, 2396)	38.1 (9.9, 59.5)	38.2 (9.6, 60.2)	0.04 (0.03, 0.05)	6 (2, 10)	5 (1, 9)	−0.7 (−0.9, −0.5)
Hungary	1734 (469, 2809)	2049 (575, 3269)	38.7 (11.1, 60.1)	39.3 (10.9, 61)	0.06 (0.05, 0.08)	12 (3, 19)	10 (3, 16)	−0.57 (−0.82, −0.330
Iceland	19 (4, 30)	27 (6, 44)	39.9 (8.6, 62.5)	39.3 (8.3, 62.2)	−0.07 (−0.08, −0.06)	6 (1, 10)	4 (1, 7)	−1.08 (−1.2, −0.96)
Ireland	398 (87, 638)	449 (95, 741)	38.1 (8.4, 60.3)	37.9 (7.6, 60.6)	0.02 (0.01, 0.03)	10 (2, 16)	5 (1, 9)	−1.7 (−1.78, −1.620
Israel	380 (97, 612)	656 (177, 1058)	37.1 (9.9, 57.9)	37.2 (9.5, 58.5)	0.025 (0.015, 0.034)	8 (2, 13)	5 (1, 8)	−2.11 (−2.41, −1.81)
Italy	7092 (1718, 10,959)	9281 (2127, 14,597)	39.9 (9.9, 62.1)	39.6 (9.4, 62)	−0.013 (−0.018, −0.007)	8 (2, 12)	6 (1, 9)	−1.11 (−1.23, −1.0)
Japan	12,011 (3491, 18,581)	26,082 (8321, 40,865)	38.3 (11.1, 59.2)	38.4 (11.9, 58.7)	0.09 (0.06, 0.12)	7 (2, 11)	6 (2, 10)	−0.54 (−0.60, −0.49)
Latvia	268 (48, 427)	291 (50, 480)	42.0 (7.5, 65.4)	41.3 (7.3, 64.6)	−0.07 (−0.09, −0.05)	7 (1, 12)	7 (1, 11)	−0.18 (−0.36, 0)
Lithuania	321 (65, 502)	434 (78, 700)	43.2 (9.0, 66.3)	41.8 (7.5, 65.7)	−0.13 (−0.14, −0.11)	7 (1, 11)	7 (1, 11)	−0.05 (−0.21, 0.10)
Luxembourg	59 (11, 92)	67 (12, 107)	40.4 (8.0, 63.4)	39.9 (7.8, 63)	−0.04 (−0.05, −0.02)	11 (2, 17)	6 (1, 9)	−1.94 (−2.12, −1.77)
Mexico	908 (332, 1378)	3886 (1303, 6170)	36.3 (13.4, 55.2)	35.6 (11.9, 55)	−0.1 (−0.12, −0.08)	2 (1, 4)	3 (1, 5)	1.03 (0.87, 1.19)
Netherlands	1850 (412, 2949)	2868 (577, 4623)	37.9 (8.6, 60.0)	38.5 (8.2, 61.0)	0.07 (0.06, 0.08)	9 (2, 14)	8 (2, 12)	−0.43 (−0.6, −0.26)
New Zealand	468 (122, 734)	662 (192, 1028)	39.9 (10.5, 62.0)	41.3 (11.8, 63.4)	0.13 (0.1, 0.16)	12 (3, 19)	8 (2, 12)	−1.58 (−1.64, −1.51)
Norway	705 (137, 1124)	820 (149, 1291)	41.1 (7.9, 64.5)	41.2 (7.7, 64.3)	0.02 (0.01, 0.03)	10 (2, 16)	7 (1, 12)	−0.96 (−1.05, −0.87)
Poland	3475 (1017, 5382)	6801 (1903, 10,813)	37.3 (11.1, 58.2)	37.4 (10.4, 58.7)	0.01 (−0.01, 0.04)	8 (2, 12)	9 (3, 14)	0.21 (0.04, 0.38)
Portugal	1185 (388, 1856)	1783 (467, 2818)	38.1 (11.8, 58.8)	38.1 (10.7, 59.6)	0.04 (0.02, 0.05)	9 (3, 14)	7 (2, 10)	−0.85 (−1.04, −0.67)
Republic of Korea	1385 (476, 2189)	4334 (1406, 7197)	36.5 (12.1, 55.7)	37.1 (11.8, 57)	0.01 (−0.04, 0.06)	5 (2, 8)	5 (2, 8)	−0.52 (−0.72, −0.32)
Slovakia	636 (193, 996)	954 (249, 1530)	39.2 (11.9, 60.5)	39.9 (11.1, 61.7)	0.08 (0.07, 0.09)	11 (3, 17)	10 (3, 16)	−0.22 (−0.32, −0.12)
Slovenia	213 (52, 339)	308 (58, 512)	40.1 (9.1, 62.9)	39.1 (8, 62)	−0.09 (−0.1, −0.08)	9 (2, 14)	6 (1, 11)	−1.18 (−1.47, −0.89)
Spain	4500 (1055, 7039)	7332 (1805, 11,511)	41.4 (10, 63.8)	40.3 (10, 62.7)	−0.07 (−0.09, −0.06)	8 (2, 13)	7 (2, 10)	−0.57 (−0.68, −0.45)
Sweden	1186 (248, 1895)	1400 (288, 2264)	39.7 (8.1, 62.8)	40.1 (7.7, 63.4)	0.051 (0.047, 0.054)	7 (2, 12)	6 (1, 9)	−0.73 (−0.89, −0.57)
Switzerland	636 (139, 1026)	821 (186, 1329)	39.1 (8.8, 61.5)	39.2 (8.7, 61.9)	0.03 (0.02, 0.04)	6 (1, 9)	4 (1, 7)	−1.26 (−1.41, −1.10)
Türkiye	1896 (644, 2988)	3957 (1236, 6428)	34.3 (11.3, 53.1)	34.1 10.7, 53.1)	−0.03 (−0.04, −0.02)	6 (2, 9)	4 (1, 7)	−0.86 (−1.2, −0.520
United Kingdom	9163 (2118, 14,306)	8801 (1806, 14,038)	40.4 (9.4, 62.9)	39.7 (8.2, 62.2)	−0.04 (−0.05, −0.02)	10 (2, 15)	6 (1, 10)	−1.48 (−1.6, −1.37)
USA	26,740 (6011, 41,721)	30,257 (6050, 47,938)	39.8 (8.9, 62.5)	40.3 (8, 63.4)	0.06 (0.05, 0.07)	8 (2, 13)	5 (1, 8)	−1.65 (−1.72, −1.58)

UI = uncertainty interval; ASDR = age-standardised death rate; EAPC = estimated annual percentage of change; OECD = Organisation for Economic Co-operation and Development, CI = confidence interval; ASPD = age-standardised percentages of death.

**Table 2 nutrients-17-01320-t002:** DALYs, ASR, ASP, and change trends of diet-related CRC between 1990 and 2021 among OECD countries.

Country	Number (95% UI)		ASP of DALYs(95% UI)		EAPC in ASP of DALYs (95% CI)	ASR of DALYs (95% UI)		EAPC in ASR of DALYs (95% CI)
	1990	2021	1990	2021	1990–2021	1990	2021	1990–2021
OECD	2,447,568 (587,458, 3,793,421)	3,088,502 (722,609, 4,844,604)	39.2 (9.3, 61.3)	39.1 (9.1, 61.3)	0.01 (0.0, 0.02)	185 (44, 286)	129 (30, 202)	−1.2 1(−1.24, −1.18)
Australia	41,810 (9654, 66,014)	53,560 (11,898, 85,652)	39.2 (8.9, 61.8)	39.7 (8.6, 62.6)	0.05 (0.04, 0.06)	217 (50, 343)	125 (28, 201)	−2.01 (−2.11, −1.9)
Austria	23,934 (5575, 38,183)	17,098 (3535, 27,380)	38.6 (8.7, 60.7)	38.6 (8.1, 61)	0.02 (0.01, 0.03)	208 (48, 334)	98 (20, 157)	−2.44 (−2.49, −2.39)
Belgium	31,156 (7250, 48,714)	27,094 (5386, 43,031)	39.1 (8.7, 61.6)	39.9 (8.1, 62.7)	0.09 (0.08, 0.100)	206 (48, 323)	120 (23, 192)	−1.66 (−1.78, −1.55)
Canada	59,373 (15,197, 92,681)	83,514 (18,914, 134,678)	39.4 (10.1, 61.4)	38.8 (9.3, 61.1)	−0.04 (−0.05, −0.03)	185 (47, 289)	123 (28, 199)	−1.12 (−1.21, −1.04)
Chile	11,988 (4091, 18,126)	31,846 (8928, 49,564)	43.2 (14.8, 64.2)	42.9 (12.5, 65.2)	0.03 (0.01, 0.05)	118 (41, 179)	126 (35, 195)	0.57 (0.43, 0.71)
Colombia	15,613 (6063, 22,799)	52,113 (17,201, 82,519)	38.3 (15.3, 57.0)	36.6 (12.8, 56.0)	−0.07 (−0.11, −0.04	84 (33, 123)	95 (31, 150)	0.31 (0.13, 0.49)
Costa Rica	1455 (502, 2191)	7248 (2230, 11,553)	37 (13, 56.3)	35.5 (11.4, 55.0)	−0.14 (−0.15, −0.13)	80 (28, 121)	132 (41, 210)	1.85 (1.63, 2.06)
Czechia	40,019 (9904, 64,933)	34,002 (8091, 56,056)	37.0 (9.3, 58.5)	36.8 (8.6, 58.7)	−0.02 (−0.03, −0.01)	292 (72, 473)	164 (38, 271)	−2.17 (−2.34, −2)
Denmark	16,097 (3603, 25,778)	17,938 (3532, 28,801)	39.1 (8.3, 61.6)	39.7 (8.0, 62.8)	0.07 (0.06, 0.07)	207 (46, 331)	153 (30, 245)	−1.29 (−1.56, −1.03)
Estonia	3749 (630, 5953)	4075 (623, 6651)	41.2 (7.3, 64.7)	40.5 (6.5, 64.2)	−0.07 (−0.08, −0.06)	183 (31, 291)	155 (23, 252)	−0.92 (−1.1, −0.75)
Finland	8961 (2028, 14,537)	11,372 (2282, 18,457)	38.7 (8.4, 61.0)	38.5 (7.9, 61.1)	−0.01 (−0.02, 0.01)	128 (29, 207)	94 (19, 152)	−0.98 (−1.06, −0.91)
France	164,856 (36,239, 257,031)	176,774 (35,873, 283,700	40.2 (8.4, 63.2)	40.5 (8.3, 63.7)	0.034 (0.029, 0.04)	204 (44, 318)	131 (26, 210)	−1.33 (−1.39, −1.28)
Germany	284,382 (60,709, 448,644)	233,980 (42,468, 377,323)	39.9 (8.2, 62.7)	39.9 (7.4, 63.3)	0.009 (0.0, 0.01)	228 (48, 359)	129 (23, 209)	−2.06 (−2.17, −1.96)
Greece	18,682 (4608, 29,483)	26,319 (5988, 42,194)	38.2 (9.6, 59.8)	38.6 (8.9, 60.7)	0.06 (0.05, 0.07)	125 (31, 197)	115 (26, 183)	−0.49 (−0.64, −0.34)
Hungary	39,246 (10,315, 63,541)	44,491 (12,098, 70,563)	38.6 (10.8, 60.2)	39.3 (10.5, 61.1)	0.08 (0.06, 0.09	271 (71, 437)	242 (65, 384)	−0.46 (−0.7, −0.22)
Iceland	398 (85, 639)	539 (109, 874)	40.2 (8.4, 62.9)	39.5 (8, 62.7)	−0.07 (−0.08, −0.06)	142 (30, 227)	95 (19, 154)	−1.26 (−1.36, −1.16)
Ireland	8631 (1819, 13,784)	9161 (1863, 15,075)	38.2 (8.2, 60.7)	38.2 (7.3, 61.1)	0.04 (0.02, 0.05)	216 (45, 345)	118 (24, 194)	−1.8 (−1.87, −1.72)
Israel	8012 (1981, 12,911)	12,317 (3310, 20,045)	37.2 (9.6, 58.1)	37.3 (9.1, 58.9)	0.03 (0.02, 0.04)	168 (41, 271)	101 (27, 165)	−2.09 (−2.38, −1.8)
Italy	152,655 (35,816, 236,320)	169,830 (38,242, 266,960)	39.9 (9.5, 62.2)	39.7 (8.8, 62.5)	−0.006 (−0.01, −0.001)	177 (41, 273)	123 (27, 192)	−1.22 (−1.36, −1.08)
Japan	290,209 (80,950, 446,618)	455,770 (134,725, 709,276)	38.5 (10.7, 59.6)	39.3 (10.9, 60.6)	0.11 (0.07, 0.14)	171 (48, 264)	141 (39, 217)	−0.67 (−0.72, −0.61)
Latvia	6464 (1114, 10,336)	5895 (961, 9689)	42.1 (7.3, 65.5)	41.4 (6.8, 65.1)	−0.06 (−0.08, −0.05)	182 (31, 291)	158 (25, 260)	−0.49 (−0.65, −0.33)
Lithuania	7731 (1499, 12,025)	8699 (1464, 13,964)	43.2 (8.6, 66.5)	41.9 (7.1, 66)	−0.12 (−0.14, −0.11	173 (36, 269)	159 (26, 255)	−0.32 (−0.49, −0.15)
Luxembourg	1259 (229, 1980)	1281 (219, 2030)	40.5 (7.6, 63.7)	40.2 (7.3, 63.7)	−0.02 (−0.04, −0.002)	236 (43, 370)	121 (20, 191)	−2.11 (−2.3, −1.91)
Mexico	23,137 (8256, 35,025)	103,481 (33,582, 165,002)	36.2 (13.1, 55.1)	35.4 (11.6, 54.8)	−0.08 (−0.09, −0.06	52 (19, 78)	79 (26, 126)	1.47 (1.33, 1.61)
Netherlands	39,915 (8615, 62,804)	58,703 (11,593, 94,146)	38.1 (8.3, 60.4)	38.8 (7.9, 61.5)	0.08 (0.06, 0.09)	203 (44, 319)	172 (34, 277)	−0.43 (−0.6, −0.25)
New Zealand	10,802 (2727, 16,969)	13,248 (3730, 20,797)	39.9 (10.1, 62.2)	41.5 (11.4, 63.8)	0.15 (0.12, 0.18	282 (71, 444)	162 (45, 255)	−1.9 (−1.97, −1.82
Norway	14,184 (2642, 22,565)	15,173 (2627, 23,707)	41.2 (7.5, 64.9)	41.5 (7.2, 64.9)	0.03 (0.025, 0.04)	219 (40, 348)	153 (26, 239)	−1.19 (−1.28, −1.09)
Poland	81,586 (23,281, 126,802)	137,297 (37,548, 218,480)	37.2 (10.8, 58.1)	37.4 (10.0, 58.8)	0.04 (0.01, 0.06)	187 (53, 290)	196 (53, 311)	0.07 (−0.1, 0.24)
Portugal	25,514 (8043, 40,163	33,503 (8236, 53,029)	38.0 (11.5, 58.8)	38.3 (9.9, 60.0)	0.07 (0.05, 0.09)	190 (60, 299)	146 (35, 232)	−0.7 (−0.93, −0.48)
Republic of Korea	39,094 (12,987, 61,733)	90,428 (27,166, 151,428	36.4 (12.0, 55.6)	36.8 (11.1, 57.0)	0.05 (0.0, 0.11)	122 (42, 193)	99 (30, 165)	−0.82 (−1.04, −0.59)
Slovakia	15,332 (4510, 23,998)	21,188 (5375, 34,094)	39.2 (11.7, 60.5)	40.0 (10.7, 61.9)	0.09 (0.08, 0.1)	256 75, 402)	225 (57, 362)	−0.4 (−0.5, −0.31)
Slovenia	4874 (1126, 7758)	5752 (1052, 9567)	40.0 (8.7, 63)	39.2 (7.5, 62.2)	−0.08 (−0.09, −0.06	198 (46, 315)	133 (24, 221)	−1.52 (−1.82, −1.22)
Spain	97,143 (21,820, 151,549)	137,666 (32,843, 216,061)	41.4 (9.5, 64)	40.4 (9.4, 63.1)	−0.07 (−0.09, −0.06)	184 (41, 288)	146 (34, 229)	−0.67 (−0.78, −0.55)
Sweden	23,510 (4742, 37,458)	24,843 (5007, 40,676)	39.7 (7.8, 62.8)	40.4 (7.3, 63.9)	0.07 (0.067, 0.08)	164 (33, 261)	119 (23, 193)	−0.85 (−1.01, −0.68)
Switzerland	13,252 (2779, 21,276)	15,328 (3353, 24,440)	39.2 (8.4, 61.7)	39.5 (8.2, 62.3)	0.04 (0.035, 0.05	132 (27, 210)	87 (19, 137)	−1.44 (−1.61, −1.27)
Türkiye	53,019 (17,530, 84,868)	95,901 (29,451, 157,132)	34.2 (11.2, 53.1)	33.8 (10.4, 52.8)	−0.03 (−0.04, −0.02)	142 (48, 226)	101 (31, 166)	−1.16 (−1.48, −0.85)
United Kingdom	188,188 (41,836, 291,875)	165,949 (32,736, 263,323)	40.4 (8.8, 63.1)	40.0 (7.7, 62.8)	−0.02 (−0.03, 0)	216 (47, 335)	134 (26, 213)	−1.58 (−1.69, −1.46)
USA	581,339 (124,030, 904,032)	685,126 (130,008, 1,077,155)	39.8 (8.4, 62.7)	40.5 (7.5, 63.9)	0.08 (0.07, 0.09)	187 (39, 290)	128 (24, 201)	−1.32 (−1.38, −1.26)

UI = uncertainty interval; ASR = age-standardised rate; EAPC = estimated annual percentage of change; DALYs = disability-adjusted life years; OECD = Organisation for Economic Co-operation and Development; CI = confidence interval; ASP DALYs = age standardised percentage of disability-adjusted life years.

## Data Availability

The data used to prepare this manuscript are freely available on the GBD website (https://ghdx.healthdata.org/gbd-2021).

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
