# Peer review of "Burden and Trends of Diet-Related Colorectal Cancer in OECD Countries: Systematic Analysis Based on Global Burden of Disease Study 1990–2021 with Projections to 2050"

_nutrients, 2025, doi:10.3390/nu17081320_

Round 1
Reviewer 1 Report
Comments and Suggestions for Authors
The manuscript titled “ nutrients-3568204_ Burden and trends of diet-related colorectal cancer in OECD countries: A systematic analysis based on Global Burden of Disease study 19990–2021 with projections to 2050“ is submitted to the Issue “Clinical Nutrition” in the Special Issue “Nutrition and Dietary Guidelines for Colorectal Cancer Patients”.
This study evaluates the diet-related burden of colorectal cancer (CRC) from 1990 to 2021 in Organisation for Economic Co-operation and Development (OECD) nations and estimates the projected burden up to 2050.
Comments
- Title: There is an error in the reference to the year 1990. Please correct it.
- Abstract: Several abbreviations are used without providing their full meaning at first mention.
- Keywords: Keywords are fundamental for article classification and searchability. Typically, words rather than phrases are used. Consider referring to the MeSH classification for guidance.
- Introduction: The introduction effectively highlights the importance of colorectal cancer, its increasing prevalence, and its relationship with diet, supported by relevant literature.
- Objective: The current wording of the objective implies a global analysis, whereas the study actually examines each of the 38 OECD countries individually. This should be revised for accuracy.
- Materials and Methods:
- The study design is not explicitly stated. Given the nature of the analysis, it appears to be an ecological trend study, where the unit of analysis is the country. This should be clearly stated.
- Abbreviations should be defined at first use; however, this is inconsistent throughout the section and should be corrected.
- The methodological approach to dietary patterns across different countries is unclear. Although Figures 1, 2, 3, and 4 present these variations, the methodology used to assess them should be explicitly described.
- The methodology for Figures 5, 6, and 7 also requires clarification.
- The study uses uncertainty intervals (UI) instead of the conventional 95% confidence intervals (CI). Could the authors justify this choice?
- In Figures 8 and 9, all abbreviations should be explained in the figure captions.
- Discussion: The discussion should begin with key findings rather than reiterating the study objective.
- Study Limitations:
- The analysis includes years affected by the COVID-19 pandemic, during which data collection was limited, even for colorectal cancer. This should be acknowledged as a potential limitation.
- Dietary trends are evolving, which may result in an underestimation of the projected impact of diet on colorectal cancer. This should be discussed.
- Conclusion: The conclusions must align strictly with the study results.
Author Response
Thank you very much for taking the time to review this manuscript. Please find the detailed responses below and the corresponding revisions in track changes in the resubmitted files.

Reviewer 2 Report
Comments and Suggestions for Authors
Our fellow researchers, with their paper, wanted to take stock of the current situation regarding colorectal cancer and project the current numbers into the near future. They correctly searched for numbers relating to practically all the countries that make their data available. The result is an analysis that sees a certain decrease in neoplasms compared to about a decade ago. In their paper they identify diet as the main moment that generates the conditions for the onset of neoplasia. We agree with this, but other elements must be taken into consideration such as intestinal dysbiosis (doi.org/10.3390/jcm13216578 to be read and cited in the bibliography). In fact, it is not only diet that can change the intestinal microclimate but also chronic therapies that for various reasons are conducted by the human race. Furthermore, a protective factor has been found in the Mediterranean diet, there are those elements that favor a perfect homeostasis of the intestinal microbiota, precisely because it is rich in waste and poor in red or processed meat, in which white meat and fish are found. We also fear that rather than a decrease in the incidence linked to the diet, the best progress has been obtained with prevention, which is recommended starting from the age of 45 with periodic checks. It is advisable to review the paper in light of these observations. Good iconography, good English, bibliography to be reviewed
Author Response

(The authors gave the same response as above.)

Round 2
Reviewer 1 Report
Comments and Suggestions for Authors
Thank you very much for allowing me to review once again the manuscript entitled “ nutrients-3568204_ Burden and trends of diet-related colorectal cancer in OECD countries: A systematic analysis based on Global Burden of Disease study 19990–2021 with projections to 2050“ as well as the authors' responses to the reviewers’ comments and suggestions, which have contributed to a clearer understanding of the work undertaken.
This study underscores the ongoing public health challenge of diet-related colorectal cancer (CRC) in OECD countries. Despite overall declines in mortality and DALYs, some nations—such as Chile, Costa Rica, Mexico, Poland, and Colombia—are facing rising trends. The burden is greater among males, with a growing gender disparity, and cases among younger adults are also increasing. Projections indicate that, although age-standardised rates may decline, the total number of diet-related CRC deaths and DALYs could continue to rise until 2050.
Having reviewed the revised version of the manuscript, I consider that all the previously raised concerns have been satisfactorily addressed.
Reviewer 2 Report
Comments and Suggestions for Authors
Our fellow researchers have made significant changes that have not distorted the initial paper but have improved it in general. The focus on diet is very clear even if we do not have at least to date the means to be able to intervene significantly on this. The result is a projection up to 2050 which is what we can read in the paper. Happy reading approved for publication